# Leave No-One Behind: A Retrospective Study of Hepatitis C Testing and Linkage to Care for Hospital Inpatients

**DOI:** 10.3390/v15040913

**Published:** 2023-03-31

**Authors:** Christine Roder, Carl Cosgrave, Kathryn Mackie, Stuart K. Roberts, Margaret E. Hellard, Amanda J. Wade, Joseph S. Doyle

**Affiliations:** 1Disease Elimination Program, Burnet Institute, Melbourne 3004, Australia; christine.roder@barwonhealth.org.au (C.R.); margaret.hellard@burnet.edu.au (M.E.H.); amanda.wade@burnet.edu.au (A.J.W.); 2Centre for Innovation in Infectious Disease and Immunology Research (CIIDIR), Deakin University, Geelong 3220, Australia; 3Barwon South West Public Health Unit, Barwon Health, Geelong 3220, Australia; carl.cosgrave@barwonhealth.org.au (C.C.); kate.mackie@barwonhealth.org.au (K.M.); 4Department of Gastroenterology, Alfred Health and Monash University, Melbourne 3004, Australia; stuart.roberts@monash.edu; 5Department of Infectious Disease, Alfred Health and Monash University, Melbourne 3004, Australia

**Keywords:** hepatitis C, testing, linkage to care, cascade of care

## Abstract

Hospital admissions are a missed opportunity to engage people living with hepatitis C virus (HCV) into care. This study aimed to describe the proportion of hospital inpatients and emergency department (ED) patients identified with hepatitis C who were subsequently linked to care and treatment at a metropolitan health service in Melbourne, Australia. Data were collected retrospectively from hospital databases (admissions, notifiable diseases, and pharmacy) for all adults admitted or attending the ED with separation coding indicating hepatitis C infection from March 2016 to March 2019. There were 2149 patients with at least one separation with hepatitis C coding. 15.4% (331/2149) had a documented antibody test, 4.6% (99/2149) had a documented RNA test, and 8.3% (179/2149) had a DAA prescription dispensed by hospital pharmacy. Antibody positivity was 95.2% (315/331) and RNA (when completed) was detected in 37.4% (37/99). Hepatitis specialist units had the highest rate of hepatitis C coded separations and RNA testing (39/88; 44.3%), mental health had the highest rate of antibody testing (70/276; 25.4%). Emergency had the lowest rate of antibody testing (101/1075; 13.7%) and the third highest rate of RNA testing (32/94; 34.1%), but the highest rate of RNA detected (15/32; 46.9%). This study highlights key steps to improve the care cascade. Simplified diagnostic pathways, expansion of hepatitis C care services, and clear in-hospital pathways to link patients to care would be beneficial in this setting. To scale up hepatitis C testing and treatment as part of national elimination strategies, hospital systems need to target interventions to their local data.

## 1. Introduction

Hepatitis C is a major, global public health problem. The World Health Organization (WHO) estimated in 2015, that 71 million people worldwide were living with hepatitis C, and the complications of untreated infection, cirrhosis, and hepatocellular carcinoma were responsible for ~400,000 deaths [1,2]. A global strategy to eliminate hepatitis C as a public health threat was announced by the WHO in 2016 [1,3]. The most recent global progress report estimates the number of people receiving treatment annually has increased from 122,000 in 2015 to >2.6 million in 2018 [2]. Despite this progress, access to diagnostics and treatment remains a barrier for many. Key priorities identified by the WHO to address these barriers include developing initiatives in line with the sustainable development goal to “leave no one behind”, as well as to “leverage synergies across the health sector” [2].

Elimination of hepatitis C is possible due to ongoing harm reduction strategies and the introduction of direct-acting antivirals (DAAs). Previously treatment with peg-interferon and ribavirin were associated with substantial side effects and a very poor sustained virologic response (SVR) of 30–50% [4]. Now, treatment has been simplified to a “treat all” approach, with the exception of patients who are pregnant or children under 12 [2]. The WHO-recommended pan-genotypic regimens sofosbuvir/velpatasvir and glecaprevir/pibrentasvir are orally administered for a short duration (8 or 12 weeks) with good tolerability and high efficacy (SVR > 90%) in real-world settings [2].

Australia is a global leader in hepatitis C elimination—in the first four years after DAA treatment was made available in 2016, before the COVID-19 pandemic, 47% of 188,951 Australians living with hepatitis C had received treatment [5]. Initiatives implemented in Australia to successfully expand access to DAAs include universal prescribing subsidised by the government (Pharmaceutical Benefits Scheme (PBS)) [4,6,7], and models of care that provided services where priority populations were more likely to access them, such as general practice, community health services, drug and alcohol services, needle and syringe programs, and mental health services [8,9,10,11].

Despite this promising start, there are still many Australians living with hepatitis C who remain undiagnosed or untreated [12]. After a sizeable initial uptake of treatment in 2016 (32,503 people treated), treatment uptake has steadily declined to just 6474 people in 2021 [13]. Innovative methods to identify people living with hepatitis C and link them to care and treatment is required for individuals to realise the benefits of treatment, and to reduce onwards transmission, thereby achieving the WHO elimination targets in Australia [3,14].

There is growing evidence that screening people at risk of hepatitis C during admission to hospital is a feasible testing strategy for inpatients [7,15,16,17]. Studies have found that with suitable linkage to care, hospital was a good setting to initiate DAAs. Although some studies trialled universal screening, the overall recommendations were to implement targeted screening for priority populations including people who inject drugs (PWID), people with mental illness, and region-specific at-risk populations such as veterans in the United States, or the 1948–1968 birth cohort in Italy [7,15,16,17].

This study aimed to describe the rates of hepatitis C testing and linkage to care among inpatients and emergency patients at a metropolitan health service in Melbourne, Australia. By determining the prevalence of hepatitis C testing and linkage to care in inpatients, this retrospective study will assess gaps in the care cascade. This critical knowledge will be used to inform new interventions and increase engagement in hepatitis C care.

## 2. Materials and Methods

### 2.1. Study Setting and Population

Alfred Health is an inner-city tertiary hospital in Melbourne, Australia, servicing a population of approximately 700,000 people in the hospital catchment area [18] defined in Table 1. Hepatitis C testing and treatment is available via the tertiary hospital and general practice.

At Alfred Health, hepatitis C is managed via specialist-run outpatient clinics located on-site, and nurse-led community-based clinics. All adult patients admitted to the health service as an inpatient or who attended the emergency department (ED) for a 3 year period (March 2016–2019) with separation coding indicative of hepatitis C as defined in Table 1 were included in the study.

### 2.2. Study Design

This study was a retrospective audit of hepatitis C testing and linkage to care for hospital patients with a history of hepatitis C. Data were obtained from hospital admissions data (all separations and hepatitis C coded separations), hospital pharmacy (dispensed DAA prescriptions), and the notifiable infectious disease database maintained by the Department of Infectious Disease (pathology data: HCV antibody and RNA testing and results). The target International Classification of Diseases, Tenth revision (ICD-10) [19] codes and other definitions used in this study are outlined in Table 1. Data collected for analysis included patient name, hospital record number, date of birth, sex, admission date, discharge date, admitted ward, ICD-10 code, pathology test date, pathology test performed (i.e., HCV antibody or HCV PCR), test result, dispensing date, and generic name of the drug.

Additional hepatitis C prevalence and treatment data were obtained from the Viral Hepatitis Mapping Project, National Report 2020 [5] to determine the number of people living with hepatitis C in the hospital catchment area.

Primary study outcomes were the proportion of hospital inpatients who:Had a separation with hepatitis C coding;Had hepatitis C virus (HCV) antibody testing (among those with hepatitis C coding) linked to admission, see below;Had HCV RNA testing (among those who were HCV antibody positive);Had evidence of treatment (among those who were HCV RNA positive and among those who had hepatitis C coding).

Secondary study outcomes:The estimated number of people living with hepatitis C in the hospital catchment area.An estimate of the number of missed opportunities in primary outcomes 2–4.

### 2.3. Data Analysis

Data management and analysis were performed using StataIC17 (College Station, TX, USA). The pathology and pharmacy data were aligned with the admissions data using identifying information and then de-identified for analysis. Descriptive analysis included count and summary statistics (means, medians, and proportions as appropriate). Patients were counted in outcome 1 if they had a separation with ICD-10 coding hepatitis C. Coding may have been from diagnosis prior to their admission or from testing that occurred as a part of their admission. Testing data (HCV antibody and HCV RNA) were counted in outcomes 2 and 3 if they occurred at any point within the study period. Evidence of treatment was counted in outcome 4 if it occurred within the study period. Logistic regression was used to analyse the relationship between admitted specialty and HCV antibody and RNA testing.

The number of people living with hepatitis C for the hospital catchment area was determined using data from the *Viral Hepatitis Mapping Project, National Report 2020* [5]. Using the proportion of people living with hepatitis C [5] in the hospital catchment area, the missed opportunities were estimated by calculating the number of people living with hepatitis C among those admitted, and then for each step of the care cascade.

## 3. Results

### 3.1. Patient Summary

During the study period, 89,852 adults (12% of the hospital catchment population of 700,000) attended the hospital as an inpatient or emergency department patient. Of these, 2149 (2.4%) had at least 1 episode with hepatitis C separation coding. There were a total of 136,319 episodes (average of 1.5 admissions per person), and 4901 had hepatitis C separation coding. The median length of stay was 2 days (IQR 1–4) for all episodes and 1.6 days (IQR 0.3–6.2) for hepatitis C coded episodes.

Overall, the average age at admission was 56 years (range 18–107), and for hepatitis C coded patients it was 50 years (range 18–93). For all episodes, 51% of patients were male, and for hepatitis C coded patients, 68% were male.

### 3.2. Hepatitis C Care Cascade

Of the 2149 patients with a hepatitis C coded episode, 331 (15.4%) had documented antibody testing: 315/331 were antibody positive (95.2%) and 16/331 were antibody negative (4.8%). RNA testing was performed in 99 patients (31.4% of antibody positive); RNA was detected in 37/99 (37.4%) and not detected in 62/99 (62.6%). There were 179 patients with a hepatitis C coded admission who had DAA prescriptions dispensed at the hospital pharmacy during the study period. Of these 5 (2.7%) had an RNA-detected test (13.5% of all RNA detected), 24 (30.4%) had an antibody-positive test with no RNA test, 9 (5%) had an RNA not detected test, and the remaining 141 (22.9%) had no testing in the hospital. The cascade of proportion of patients with hepatitis C, to testing and treatment is shown in Figure 1.

Hepatitis specialist units had the highest number of episodes with hepatitis C separation coding with 1159 episodes for 576 patients. Patients with episodes in mental health had the highest rates of antibody testing (70/276, 25.4%), while patients with episodes in emergency had the lowest (101/738, 13.7%). Patients with episodes in hepatitis specialist units had the highest rates of RNA testing (39/88, 44.3%), while patients with episodes in general medicine had the lowest (15/59, 25.4%). The cascade of care per specialty is shown in Figure 2 and Table 2.

The unadjusted odds of RNA testing (Table 3) increased significantly among patients with episodes in hepatitis specialist units (OR 1.95, 95% CI 1.26–3.02) and mental health (OR 2.22. 95% CI 1.34–3.69) compared with patients who did not have episodes in these specialties (OR 0.03, 95% CI 0.03–0.05 and OR 0.04, 95% CI 0.03–0.05). When adjusted for age and sex there were no significant associations between specialty and RNA testing. There was a significant association between male sex and an increase in both antibody testing (OR 2.71, 95% CI 2.29–3.20) and RNA testing (OR 3.04, 95% CI 2.29–4.03).

### 3.3. Identifying Missed Opportunities

In the hepatitis C care cascade above there were 1818/2149 (84.6%) people with a hepatitis C coded separation who did not have an antibody test ordered by the hospital. More importantly, there were 216/315 (68.6%) people who had a positive antibody test but did not have an RNA test ordered by the hospital.

According to the Viral Hepatitis Mapping Project, National Report 2020, the hospital catchment area for the Alfred Hospital had an estimated 5091 people living with chronic hepatitis C as of 2016 (0.71% of the total population), with 56.5% of those people treated from 2016 to 2020 [5]. Based on these data, it can be estimated that 635/89,852 (0.71%) people admitted were living with hepatitis C, and at the end of the study period 276 people (43.4%) would have remained untreated. This suggests there are potentially an additional 320 people who are HCV antibody positive and 536 people with detectable RNA to those captured in this study.

## 4. Discussion

The primary aim of this study was to measure the care cascade among hospital inpatients with a medical history of hepatitis C, particularly the step between an antibody-positive result and RNA testing that is crucial for diagnosis. This study demonstrated that approximately 2% of patients admitted to the hospital as an inpatient or attending ED had separation coding indicating a history of hepatitis C. This is already substantially higher than the 0.5% prevalence of hepatitis C in the general population of Australia, but the true proportion living with hepatitis and remaining undiagnosed may be much higher. This highlights the opportunity hospitalisation may represent to engage people in hepatitis C RNA testing, and linkage to care. Data-driven interventions are required to maximise the benefits of DAA treatment and reach elimination—and RNA testing and linkage to care whilst an inpatient may engage people currently not being reached by standard models of care.

Hepatitis specialty units had the highest rates of inpatient follow-up RNA testing of individuals with antibody positivity at 44.3%—albeit this is still less than half of individuals coded with HCV. Whilst some people may have had their RNA test in the community or outside pathology service, loss to follow-up at this point in the care cascade has been well demonstrated previously [20]. A failure to perform HCV RNA testing limits onward referral to treatment service. Clinical units with high rates of antibody screening or people diagnosed with hepatitis C, but low rates of RNA testing could benefit from a simplified diagnostic process. Models that simplify the diagnostic process could include a combination of rapid point-of-care testing, or RNA reflex testing for any laboratory HCV antibody detection [7,16,21]. One local trial of rapid antibody testing in Eds found that although it was a successful screening tool, there were similarly a very low number of people completing RNA testing and linkage to care [21]. Reflexive RNA testing on any positive serum antibody test would increase engagement in the care cascade in all units.

Low treatment uptake has been reported in other studies that investigated prevalence of hepatitis C in hospital settings. Most studies recommended commencing DAA therapy whilst the patient was still an inpatient [7,16] and show that is feasible when referred to specialty units [22]. Unfortunately, this is not simple in the current Australian regulatory environment for government-subsidised medication, and in other jurisdictions where insurance payments for medications impose prescribing barriers. Until such regulations change, inpatients could benefit from collaborative pathways linking EDs, mental health, and other services to outpatient hepatitis C treatment teams. For example, a nurse-coordinated models of care, similar to that implemented in primary care and community clinics in Melbourne [10], or a collaborative referral pathway, similar to one implemented in regional Victoria to improve the management of women with hepatitis B during pregnancy [23]. The strengths of these pathways were that they were clear and had healthcare professionals with a dedicated role in guiding their patients through the hepatitis C care cascade, reducing the number of patients lost in the process [10,23].

People with mental illness are overrepresented among those living with hepatitis C [11]. In this study, the mental health unit already had high rates of antibody testing, but there were many missed opportunities for diagnosis and treatment. To overcome these missed opportunities, health services provided by mental health units can be expanded to include hepatitis C care. This could be achieved in a similar manner to community-based mental health services that have incorporated hepatitis C care including the co-location of hepatitis C nurses and consultation pathways with specialist clinicians [9,11]. Mental health services are increasingly being identified as a setting where hepatitis C care can be incorporated to increase hepatitis C diagnosis and treatment in an environment where patients feel safe, supported, and free from the stigma they may experience elsewhere [9,11,17].

This study has some limitations. First, it specifically limited itself to hepatitis C coded separations so we cannot comment on the total proportion of people living with hepatitis C, or testing trends or risk behaviours across the broader health service population. The rationale was to quantify how many missed opportunities currently occur. Although those who have no hepatitis C history in terms of testing or diagnosis were not included in this study, people with a risk factor for hepatitis C such as injecting drug use or incarceration could be prioritised for any testing strategy implemented as a result of this study. Second, testing ordered outside hospital or ED admissions, including hepatitis C outpatient services, were not included in the pathology dataset. This means that antibody and RNA testing may have been performed after discharge, and is under-reported in this study. Nevertheless, the hospital record system did not record data on their testing at the time of retrospective review, indicating another missed opportunity to ensure treatment plans are in place. Due to prescribing requirements, we can assume that everyone who received treatment was RNA positive; however, antibody-positive numbers are still greater than treatment uptake numbers, indicating there are still many who are not getting RNA tests and then commencing treatment. Third, prescribing may have taken place in community or primary care settings so prescription data may be missing. Although this data would have been informative, the aim of the study was to understand the testing and treatment practices and identify areas where testing has occurred without follow-up within the hospital setting. Any outside treatment still appears as a missed opportunity to “close the loop”, document outcome, and update clinical coding records for the future; none of which occur currently when people are treated externally. Finally, data from 2020–2022 was not included due to the SARS-CoV-2 pandemic changing the nature of hospital admission and testing. During 2020 and 2021, the significant impact of SARS-CoV-2 restrictions on the general population and health care settings reduced accessibility to health services for many people, including those living with hepatitis C [24], and argues further for opportunistic linkage to care in the post-COVID-19 pandemic environment when possible.

## 5. Conclusions

Increasing RNA testing and linkage to care among hospital inpatients with hepatitis C will maximise the benefits of DAA and achieving elimination. Hospital admissions provide a valuable opportunity to engage people living with hepatitis C care which, if seized, will assist in accelerating progress towards elimination of hepatitis C. To maximise effectiveness and ensure feasibility, strategies for testing may need to be tailored to specific units within the hospital based on the prevalence of at-risk populations, awareness and priority among medical staff, and typical duration of stay. In addition to these, strategies for increased treatment and linkage to care will need to take into account patient-centric factors such as the type of service they can access or are currently engaged with, and the patients’ additional needs or competing health and other psychosocial interests.

## Figures and Tables

**Figure 1 viruses-15-00913-f001:**
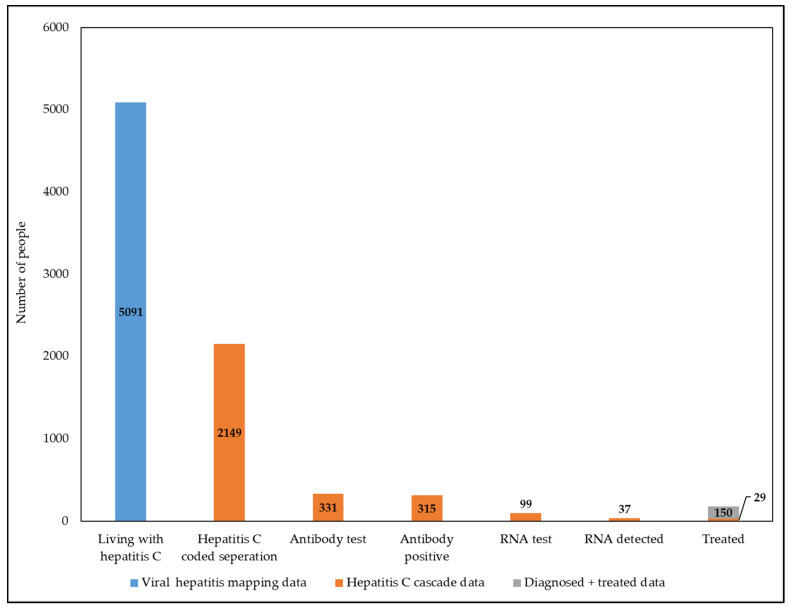
Hepatitis C care cascade for patients with a hepatitis C coded episode at the Alfred Hospital from 2016 to 2019. Viral Hepatitis Mapping Project, National Report 2020 data [5] were used to estimate the number of people living with hepatitis C in the hospitals catchment area (blue). The care cascade determined using the hospitals datasets (admissions, pathology, and pharmacy) is shown in orange. The orange portion of the treated column shows patients who also had an antibody- and/or RNA-positive test, while the grey portion shows patients with a hepatitis C coded separation and evidence of treatment.

**Figure 2 viruses-15-00913-f002:**
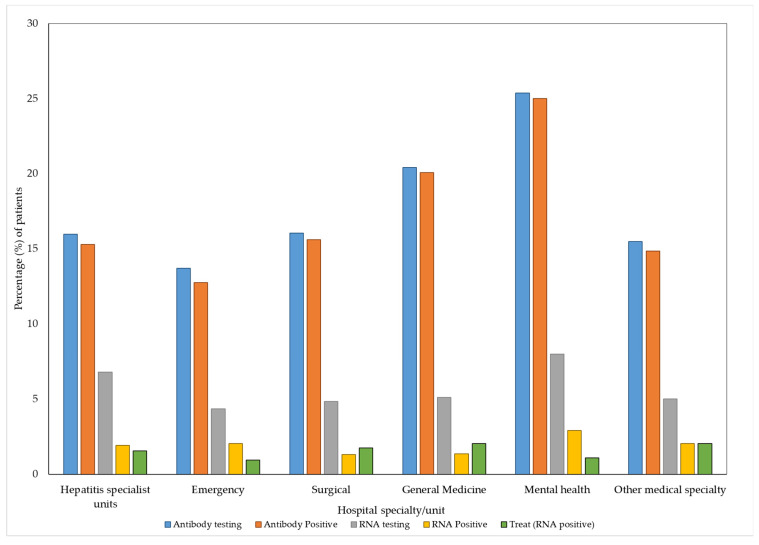
Hepatitis C care cascade for the Alfred Hospital 2016–2019 by specialty unit. The denominator (*n*) for all steps in the cascade is the patients with hepatitis C coded admissions in that specialty (Table 2). Treated includes patients who had a positive antibody or RNA test.

**Table 1 viruses-15-00913-t001:** Study definitions.

Term	Definition
Episode	Inpatient admission or attendance to emergency department (ED)
Separation coding	The diagnosis coding according to the International Classification of Diseases, Tenth revision (ICD-10) [19] as per the medical information included in the patients discharge/ED summary
Hepatitis C exposure	A positive HCV antibody test or hepatitis C separation coding
Diagnosis of hepatitis C	A positive HCV RNA test
Evidence of treatment	A prescription for DAAs dispensed at the Alfred Hospital pharmacy
Episode specialty	Admission ward
Hepatitis specialty units	Infectious disease and gastroenterology
Mental health	All psychiatric wards
Surgical	All general and specialist surgical units.
Emergency	Emergency department and trauma unit
General medicine	All general medicine wards
Other medical specialty	All non-surgical specialty wards
Hepatitis C ICD-10 codes [19]	Acute hepatitis C (B171) Chronic viral hepatitis C (B182) Chronic viral hepatitis Unspecified (B189) Unspecified viral hepatitis without hepatic coma (B199)
Hospital catchment area [18]	The combination of the following statistical area 3 (SA3) regions: Port Phillip Stonnington—West Stonnington—East Melbourne City Kingston Bayside Glen Eira

**Table 2 viruses-15-00913-t002:** The hepatitis C care cascade by specialty.

Unit	Admissions (*n* = 4901)	Unique Patients (*n* = 2149 *)	Antibody Testing, *n* (% of Patients)	Antibody Positive, *n* (% of Antibody Testing)	RNA Testing, *n* (% of Antibody Positive)	RNA Positive, *n* (% of RNA Testing)	Treatment, *n* (% of Ab Positive)	Treatment, *n* (% of RNA Positive)	Treatment, *n* (% of Unique Patients)
Hepatitis Specialist	1159 (23.6%)	576 (26.8%)	92 (16%)	88 (95.7%)	39 (44.3%)	11 (28.2%)	7 (8%)	2 (18.2%)	74 (12.9%)
Emergency	1075 (22%)	738 (34.3%)	101 (13.7%)	94 (93.1%)	32 (34.1%)	15 (46.9%)	5 (5.3%)	2 (13.3%)	61 (8.3%)
Surgical	592 (12.1%)	455 (21.2%)	73 (16%)	71 (97.3%)	22 (31%)	6 (27.3%)	7 (9.9%)	1 (16.7%)	41 (9%)
General Medicine	469 (9.6%)	294 (13.7%)	60 (20.4%)	59 (98.3%)	15 (25.4%)	4 (26.7%)	6 (10.2%)	0	32 (10.9%)
Mental Health	477 (9.7%)	276 (12.8%)	70 (25.4%)	69 (98.6%)	22 (31.9%)	8 (36.4%)	3 (4.3%)	0	39 (14.1%)
Other specialty medicine	1129 (23%)	640 (29.8%)	100 (15.7%)	95 (96%)	32 (35.7%)	13 (40.6%)	11 (11.6%)	2 (15.4%)	68 (10.6%)

* Patients may have been admitted to multiple units during the study period, *n* = 2149 applies to each row individually but not the entire column collectively.

**Table 3 viruses-15-00913-t003:** Odds ratios for patients having RNA testing by specialty.

Specialty	RNA Testing			
	OR	95% CI	aOR	95% CI
Hepatitis Specialist	1.95	1.26–3.02	1.66	0.40–6.95
Emergency	0.94	0.59–1.48	1.19	0.28–5.10
Surgical	1.18	0.71–1.94	3.58	0.42–30.79
General Medicine	1.2	0.67–2.16	1.57	0.18–13.89
Mental Health	2.22	1.34–3.69	3.16	0.37–27.39
Other medical specialties	1.05	0.66–1.67	0.51	0.13–1.98
Age	1.02	0.96–1.07		
Sex (male)	3.04	2.29–4.03		

## Data Availability

The data used for this study were individual patient data. As such they have not been made available to protect the privacy of the patients involved. Aggregated data extracts may be available on request.

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
