# Peer review of "Leave No-One Behind: A Retrospective Study of Hepatitis C Testing and Linkage to Care for Hospital Inpatients"

_viruses, 2023, doi:10.3390/v15040913_

Round 1

Reviewer 1 Report

Dear Authors,
HCV is a global health problem, the implementation of HCV screening is of paramount importance worldwide. This paper highlight the necessity to
link the hepatitis C test to care for hospital inpatients. However,
the main text must be improved. I have some comments:
1) Introduction section: add more information about DAA therapy (pangenotypic, treatment efficacy, etc),and about HCV global landascape in view of eradication by 2030 (WHO).
2) Materials and Methods: it is necessary to organize this section in at
least 3 sub-sections, such as study population, study design (maybe a schematic workflow) and data analysis. Please, put the “The following definitions were used in this study” in a table.

3) Results section: The results need to be presented more clearly, for example in figure 1 all histograms describe numbers.... it is not necessary to indicate in yellow "number" for "living" and "admitted", rather it is important to indicate the reported group.

4) Discussion section need to improved according to more recent literature in the same field.

Kind regards.

Author Response

I would like to thank reviewer 1 for taking the time to review this paper and provide constructive feedback regarding improvements. The specific points are addressed below:

1) We have included two paragraphs at the beginning of the introduction to provide an overview of the global landscape and DAA therapy. Editors please note that this has brought the word count of the article up too.

2) The materials and methods section has been restructured. Subsections include Study setting and population, study design and data analysis. We have also added more detail to the methods for clarity and the study definitions are now presented in table 1.

3) Figure 1 has been revised to present a clearer cascade of care, including changes to the legends and the figure caption.

4) The cited literature used includes papers from 2018 to 2022, which would be considered recent and up to date. There has been one additional paper cited. Without specific suggestions its difficult to make more substantial changes.

Reviewer 2 Report

Thank you for the opportunity to review this manuscript.  The authors make a compelling argument that to achieve HCV elimination better engagement of hospitalized patients needs to occur.  However the focus on only a population that 'has a coded HCV diagnosis' already seem to miss the opportunity to discuss potential role for expanded screening.

The take home message I get from the study is that patient who are known to have HCV that are admitted to the hospital are not moved across the care continuum very well.  

The only major issue with the study is the focus on those already diagnosed with HCV - or already coded.  One of the main arguments if for reflex confirmatory testing - but for this population we essentially know they are antibody positive.... (assuming coding is correct) so why do we even need to repeat the HCV Ab testing?  Should the study focus more on the population newly found to be HCV Ab+?  

- line 148 - calculation is incorrect 15/59 does not = 2.4%

- it would be interesting to know if all mental health hospitalizations and ED visits resulted in blood draws.  Often ED visits for instance (for sutures or other small issue) don't have blood draws - so it would seem inappropriate to expect HCV testing at those sort of visit.  Do you know what percentage of 738 ED visits had blood collected?  that may be a more appropriate denominator unless HCV capillary blood testing is available.

- I could consider maintaining consistency on the cascade figures - specifically related to how treatment is calculated.  In the current form is looks like more patients are being treated than are HCV RNA positive.

 - line 172-173 - I would imagine many of the 1818 who did not have repeat HCV ab testing did not actually need HCV Ab testing.  This could include patients who are currently on treatment, reported already being cured of HCV, etc.

- I didn't see a mention of how long participants follow-up was.  Did the treatment need to occur within a certain time window related to ED visit (ie within 6-, 12- months) or anytime in the study window?

Author Response

I would like to thank reviewer 2 for taking the time to review this article and provide constructive feedback regarding its improvement. The specific points/concerns have been addressed below.

1) The purpose of this paper was to identify what was happening within the hospital for patients that already have some kind of history with hepatitis C, this would allow us to identify service gaps and missed opportunities. We felt that the pathways to care within the system need to be put in place before expanding screening to capture those without a diagnosis.

2) The limitation of hepatitis C coded separations has already been addressed in the discussion and was intentional for the purpose of this study. For this specific population, yes you could make the argument that antibody testing is not necessary, however, there will be people with hepatitis C and no history of testing who come through the same setting. Ideally, hepatitis C care pathways within the hospital need to be applicable to both, and should be simplified to ensure they are incorporated into routine care.

3) Thank you for spotting this typo. The calculation should be 15/59=25/4. This has been corrected in the text.

4) This would be absolutely fascinating data but is outside of the scope of this current study. Future studies, particularly ones focusing on a single unit/department/specialty, may be able to incorporate data at this granular level. Thank you for the suggestion.

4) Figure 1 has been revised to be clearer. The treatment data in both has been revised to a) be more consistent with each other and b) provide a clearer picture of patients in the cascade. 

5) Yes I agree that some of these patients may not have required antibody testing. However, there is no evidence of external testing or treatment within the hospital records, so we cannot make the assumption that any particular patient did not require antibody testing. In the discussion, we explore the limitation of external testing and treatment and have added to this to clarify the role of documenting outcomes and updating clinical records (see lines 278-280)

6) More detail has been included in the methods to clarify this.

Round 2

Reviewer 1 Report

Dear Authors,

I agree to publish the paper in present form.

Best regards.